# Metabolomics Reveals Glycerophospholipids, Peptides, and Flavonoids Contributing to Breast Meat Flavor and Benefit Properties of Beijing-You Chicken

**DOI:** 10.3390/foods13162549

**Published:** 2024-08-15

**Authors:** Jian Zhang, Xia Chen, Jing Cao, Ailian Geng, Qin Chu, Zhixun Yan, Yao Zhang, Huagui Liu

**Affiliations:** Institute of Animal Husbandry and Veterinary Medicine, Beijing Academy of Agriculture and Forestry Sciences, Beijing 100097, China; zjcau@126.com (J.Z.); chenxia_91@163.com (X.C.); caojing2046555@163.com (J.C.); ailiangengcau@126.com (A.G.); chuqinsd@163.com (Q.C.); yanzhixun2008@sina.com (Z.Y.); duguyimeng1@126.com (Y.Z.)

**Keywords:** Beijing-You chicken, metabolomics, glycerophospholipids, peptides, flavonoids, breast meat

## Abstract

Unique metabolites contribute to the performance of meat flavor and potential function. In this study, UHPLC-Q Exactive HF-X-based metabolomics and multivariate analysis were applied to explore the characteristic metabolites in the breast meat of Beijing-You chicken (BYC) aged 150, 300, and 450 days (D150, D300, and D450). Based on the criteria of variable importance in the projection (VIP) > 1 and *p* < 0.05, a total of 154 and 97 differential metabolites (DMs) were screened out compared with D450 (D450 vs. D150, D450 vs. D300), respectively. In general, the relative content of carnosine, L-L-homoglutathione, demethyloleuropein, neohesperidin dihydrochalcone, 7-chloro-2-(3,4-dimethoxyphenyl)-3,5-dihydroxy-6,8-dimethoxy-4H-chromen-4-one, glycerophospholipids, exhibited the highest abundance at D450, while balenine, anserine, L-beta-aspartyl-L-leucine, glutathione, oxidized glutathione, stearoylcarnitine, ganoderic acid alpha, oleuroside, Lysoglycerophospholipid species (LGP) presented a downward trend with age. These 210 DMs were involved in 10 significantly enriched pathways related to the synthesis and metabolism of amino acids, peptides, and glycerophospholipid, such as glutathione metabolism, histidine metabolism, glycerophospholipid metabolism, arginine biosynthesis, tyrosine metabolism, and lysine degradation. In conclusion, this work could not only facilitate a better understanding of the differences of chicken flavor and benefit properties with age, but also provide potential valuable bioactive compounds for further research.

## 1. Introduction

Foods that contain special quality nutrients/bioactive compounds continues to attract significant attention in recent scientific studies, and are referred to as functional foods due to their health and wellbeing characteristics [1]. In the face of increasing demand for functional products, there is a great need to identify and evaluate bioactive compounds in food.

Most of the available evidence suggests that poultry meat represents a safe, low-fat source of high-quality dietary protein, which could provide the nutrients humans need [2]. Chicken, especially older indigenous native breeds, is one of the most important sources of functional food containing a wide range of bioactive compounds including carnosine (Car), anserine (Ans), balenine (Bal), glutathione, and a series of lipid molecular species [3]. For example, Silkie chicken, a famous Chinese indigenous chicken breed with prosperous medicinal and nutritional value, was referred to as a functional food product in China [4,5].

In recent years, with the development of metabolite separation, bioinformatics platforms, and mass-spectrometry resolution, the metabolomics approach has been comprehensively applied not only to characterize bioactive compounds, flavor, and nutrients in meat, particularly in chicken [6,7], but also to comprehensively understand the dynamic biochemical changes in metabolites caused by specific factors, such as animal age [8], animal genetic background [9], postmortem aging [10], and meat treatment processing [11]. A series of metabolites, such as estradiol, lipid molecules, and fatty acids, have been identified as characteristic bioactive compounds responsible for the health-promoting effects of Silkie chicken [5]. Using HPLC–QTRAP–MS-based metabolomics, the metabolites in breast meat of Beijing-You chicken are affected by age, and arginine biosynthesis, purine metabolism, alanine, aspartic acid, and glutamic acid metabolism are further suggested as important metabolic pathways [12]. Moreover, chickens with older age are highly preferred over younger ones by consumers due to their special bioactive compounds, and significant health benefits, which are widely accepted in Chinese culture [13,14]. Revealing these functional metabolites of chickens of different ages is thus fascinating for the potential health benefits of chickens with older age to be further realized, whereas studies focusing on the metabolic profiling of chickens associated with age have not yet been thoroughly studied, particularly for indigenous native breeds with age prolonged more than 450 days.

Therefore, this study selected Beijing-You chicken (BYC), a famous indigenous chicken breed with excellent meat quality, as a model system to explore the dynamic differences in metabolic profiling in breast meat throughout a wide age spectrum for a duration of 450 days through LC–MS/MS analysis strategies. Characterization and discrimination of metabolic profiles of BYC breast meat between different ages were conducted. Moreover, based on chicken at 450 days, the potential pathways and bioactive compounds of lipids, peptides, and flavonoids were further provided. This study provides an important foundation to dissect the mechanisms involved in dynamic alternations of bioactive compounds and pathways in BYC breast meat due to age.

## 2. Materials and Methods

### 2.1. Ethics Statement

The experimental procedures and animal welfare practices were conducted in accordance with the guidelines for experimental animals established by the Ministry of Science and Technology (Beijing, China). The experiment was approved by the Institute of Animal Husbandry and Veterinary Medicine, Beijing Academy of Agriculture and Forestry Sciences (Beijing, China) (No. BAAFS-IAHVM20190115).

### 2.2. Animals and Samples Collection

The detailed information on animals and sample preparation was consistent with that of our previous studies [15,16]. In brief, a total of 90 1-day-old female BYC birds with the same genetic backgrounds were provided by the Institute of Animal Husbandry and Veterinary Medicine, Beijing Academy of Agriculture and Forestry Sciences. These birds were evenly divided into three groups and raised under identical environmental and nutritional conditions with free access to feed and water throughout the whole rearing period. At each sampling age (D150, D300, and D450), 15 birds were randomly selected from each group, electrically stunned and killed by exsanguination, respectively. The samples from the left fillet (pectoralis major) were collected, snap-frozen in liquid nitrogen, and stored at −80 °C until UHPLC-Q Exactive HF-X metabolomics analysis was carried out. On the other hand, the right fillet samples were stored at 4 °C for measuring of meat quality characteristics.

### 2.3. Meat Quality Characteristics

The meat dry matter content was determined by the relative weight difference before and after drying using a freeze-dryer (Millrock Technology, Kingston, NY, USA). Then, about 5 g dried meat powder per sample was used to calculate IMF using Soxhlet extraction with petroleum ether according to the method of Zerehdaran et al. [17]. In addition, 1 g degreased meat lyophilized powder was used to obtain the composition of free amino acids and some peptides, as described by Chen et al. [18], using an amino acid analyzer (L-8900, Hitachi Ltd., Tokyo, Japan).

### 2.4. Sample Preparation of Metabolomics Study

The extraction method of metabolites was adopted according to the report by Li et al. [19]. In brief, 400 µL of methanol/water = 4:1 (*v/v*) mixture, containing 0.02 mg/mL of internal standard (L-2-chlorophenylalanine), was used to extract metabolites from the meat samples (50 mg) in a 2 mL centrifuge tube with a 6 mm diameter grinding ball. A Wonbio-96c frozen tissue grinder (Shanghai Wando Biotechnology co., LTD, Shanghai, China) was applied for 6 min (−10 °C at 50 Hz) to grind the samples, and then they were low-temperature ultrasonic extracted for 30 min (5 °C at 40 kHz). Subsequently, the samples were placed at −20 °C for 30 min, and centrifuged at 13,000× *g* for 15 min at 4 °C. Finally, the supernatant was transferred into an injection vial for LC–MS/MS analysis. Moreover, to ensure the reliability and stability of the analysis, a quality control (QC) sample was inserted into every 5–15 analysis samples. QC samples were prepared by mixing 20 µL extracts from each sample. Both the injection and detection methods of the QC samples are consistent with those of the normal samples.

### 2.5. UHPLC–MS/MS Analysis

A Thermos UHPLC-Q Exactive HF-X system equipped with an ACQUITY HSS T3 column (100 mm × 2.1 mm i.d. × 1.8 µm; Waters, Milford, CT, USA) was applied for the metabolomics profiling analysis in positive mode and negative mode. The sample extracts were separated with solvent A (0.1% formic acid in water/acetonitrile = 95:5 (*v*/*v*)) and solvent B (0.1% formic acid in acetonitrile/isopropanol/water = 47.5:47.5:5 (*v*/*v*)). The injection volume was 2 µL, and the column temperature was set at 40 °C. The data-dependent acquisition mode (DDA) was applied for data acquisition. Top 10 mother ions were collected by DDA. Before sample tested, based on the operational manual of Thermofisher Orbitrap, the instrument was calibrated with a blank sample (pure water) to identify and correct the deviation in the experimental operation.

The experimental parameters were as follows: the temperatures of ion source and capillary were set to 425 °C and 325 °C, respectively. The normalized collision energy was set to 20–40–60 V rolling for MS/MS. The spray voltage was set to (+) 3500 and (−) 3500 V. The resolutions of MS spectra and MS/MS were 60,000 and 7500 over a mass range of 70–1050 m/z, respectively. In addition, the gas flow rate of sheath and Aux were set at 50 arb and 13 arb, respectively.

### 2.6. Data Statistics and Bioinformatics Analysis

The software of Progenesis QI 2.3 (Waters Corporation, Milford, CT, USA) was applied to pretreat the LC/MS raw data. After the peaks of internal standard, noise, column bleed, and derivative reagent were removed from the three-dimensional data matrix, the metabolites were identified by searching the main database of HMDB (http://www.hmdb.ca/), Metlin (https://metlin.scripps.edu/) and the Majorbio Database. In brief, the mass tolerance between the measured m/z values and the exact mass of the components of interest was ±10 ppm. Metabolites with an MS/MS fragments score above 30 were considered as confidently identified. After searching the database, the data matrix was uploaded to Majorbio Cloud Platform (www.majorbio.com) for further data analysis through the “ropls” (Version 1.6.2) R package. Subsequently, the final normalized data matrix was obtained after filtering and normalization. Partial least squares discriminant analysis (PLS-DA) was applied to obtain an overview of the metabolic data, general clustering, trends, and visualized outliers. In addition, orthogonal partial least squares discriminant analysis (OPLS-DA) was adopted to detect the global metabolic changes between different comparable groups and provide the variable importance in the projection (VIP) for the model. In addition, 7-cycle interactive validation and the Student’s *t*-test were performed. The model parameters of R2 and Q2 were employed to evaluate the model validity avoiding the risk of over-fitting. To explore the changes in metabolite composition compared with D450, two different comparisons of metabolite composition for D450 vs. D150 and D450 vs. D300 were performed, respectively. The metabolites with the variable importance in the projection (VIP) score > 1, obtained by the OPLS-DA model, and *p*-value of *t*-test < 0.05 were identified as differential metabolites (DMs). The receiver operating characteristic (ROC) curves were determined and the under the ROC curve (AUC) was calculated to detect the efficiency of these DMs in distinguishing meat from different developmental stages. Kyoto Encyclopedia of Genes and Genomes (KEGG) metabolic pathway analysis and metabolic pathway analysis (MetPA) were adopted to further evaluate these DMs. The –log(p) values and pathway impact values of all matched pathways were visualized by plotting on the *Y*-axis and *X*-axis, respectively. In addition, data on meat quality characteristics were analyzed by the general linear model procedure of SAS (version 9.2, SAS Institute Inc., Cary, NC, USA), with the factor of age being set as the main effect. Tukey’s method was applied to identify significant differences between LSmeans (*p* < 0.05).

## 3. Results

### 3.1. Characteristics of Meat Quality during the Developmental Process

The birds’ live body weight and breast meat quality characteristics of intramuscular fat (IMF), dry matter, peptides, and free amino acids (FAAs) of BYC at three different stages are shown in Table 1. The live body weight and dry matter increased significantly with age (*p* < 0.05). As expected, IMF of D150 was lower (*p* < 0.05) than either D300 or D450 samples, which did not differ from each other (*p* > 0.05). Carnosine presented the highest at D450, followed by D300 and then D150 (*p* < 0.05). However, anserine displayed the opposite trend. Regarding the FAAs, 17 out of 19 FAAs presented significant differences (*p* < 0.05) at various stages, except for serine and proline (*p* > 0.05). Furthermore, total free amino acids at D150 were lower than those at either D300 or D450, which did not differ from each other (*p* > 0.05).

### 3.2. Metabolic Profiling and PLS-DA Results

UHPLC-Q Exactive HF-X metabolomics was applied to address the changes in the breast muscle metabolic profiles of BYC at 150, 300, and 450 days (D150, D300, and D450), during the developmental process. All metabolites identified by an ion mode of positive and negative were integrated for the following analysis. A total of 536 metabolites were detected (Appendix A), including 73 metabolite classes according to the human metabolome database (HMDB), such as carboxylic acids and derivatives, glycerophospholipids, fatty acyls, organooxygen compounds, prenol lipids, steroids and steroid derivatives (Appendix A). Thereinto, carboxylic acids and derivatives, glycerophospholipids, and fatty acyls were the top three classes in the volume of metabolites.

These metabolites of BYC during the developmental process were further analyzed using PLS-DA, a multivariate statistical analysis method, with supervised pattern recognition. A clear separation between the D150, D300, and D450 stages was observed (Appendix A), indicating that the metabolic profiles of breast muscle varied with age. Furthermore, the parameters of R2Y (0.937) and Q2 (0.799) were all more than 0.50, demonstrating that the model exhibited both good cumulative interpretation ability and strong predictive ability. Moreover, the negative intercept of Q2 regression line and the red line (R2) was always higher than the blue line (Q2), further validating the reliability of these models (Appendix A).

### 3.3. Identification of Characteristic Metabolites

To elucidate the changes in metabolites relative to D450 of BYC breast muscle, differential metabolite analyses were explored in two comparisons between D450 and D150 (D450 vs. D150), as well as between D450 and D300 (D450 vs. D300). Based on the criteria of VIP > 1 and *p* < 0.05, a total of 154 and 97 metabolites were screened out, including 44 and 47 up-regulated and 110 and 50 down-regulated in the comparisons of D450 vs. D150 and D450 vs. D300, respectively (Appendix A). These differential metabolites (DMs) were further visualized by volcano plots (Figure 1). Additionally, 41 common different metabolites were detected between these two comparisons by a Venn plot analysis (Appendix A). Ultimately, 210 unique metabolites were further identified and classified according to the HMDB database (Appendix A). These 210 differentially accumulated metabolites were predominantly categorized as carboxylic acids and derivatives (48, 25.13%), glycerophospholipids (42, 21.99%), fatty acyls (18, 9.42%), organooxygen compounds (11, 5.76%), prenol lipids (8, 4.19%), steroid and steroid derivatives (5, 2.62%), peptidomimetics (5, 2.62%) and flavonoids (4, 2.09%) (Appendix A). According to the weighed coefficients of the OPLS-DA model, the top 20 DMs were further screened out for the comparisons of D450 vs. D150 and D450 vs. D300, respectively (Figure 2). These DMs, with VIP scores more than 2, were considered as biomarkers associated with meat quality for further study.

In addition, to further study the classification and changing trend in DMs during developmental progress, the expression pattern of metabolites classified as lipids and lipid-like molecules, organic acids and derivatives, organooxygen compounds, and flavonoids, were further visualized through hierarchical clustering and heatmap analyses (Figure 3, Figure 4 and Figure 5).

#### 3.3.1. Lipids and Lipid-like Molecules

In this study, lipids and lipid-like molecules were the most abundant metabolites. A total of 73 lipids and lipid-like molecules were identified, comprising 42 glycerophospholipids, 18 fatty acyls, 8 prenol lipids, as well as 5 steroids and steroid derivatives (Appendix A). The most abundant lipid and lipid-like molecules in BYC breast meat are glycerophospholipids, containing five subclasses including phosphocholines (PC, 20 species), phosphoethanolamines (PE, 13 species), phosphates (4 species), phosphoserines (PS, 3 species), and phosphoinositols (PI, 2 species). Additionally, the expression patterns of those 73 lipids and lipid-like molecules from D150 to D450 are shown in Figure 3. On the whole, the expression patterns of lipid-related DMs at D150 were more distinguished from those at D300 and D450, which agreed with the IMF results of this study (Table 1). In addition, these DMs could be further assigned into three distinct subclusters based on the K-means clustering result during the developmental process. Subcluster 1 (the largest subcluster, 39 DMs), including LPC (18:1), Lysoglycerophospholipid species (LGP), glycerophosphate, eicosapentaenoic acid, 1-oleoylglycerophosphoserine, stearoylcarnitine, ganoderic acid alpha, oleuroside and methyl hydrogen fumarate, presented a consistent down-regulation trend from D150 to D450. However, the opposite pattern was observed in subcluster 2 (20 DMs), such as in glycerophospholipids (PC, PE, PS, and PA), aeglin, and boviquinone 4. In addition, the relative levels of 14 DMs in subcluster 3, including stearaldehyde, and some glycerophospholipids (PC and PE), decreased to the minimums at D300 and then increased. The above results suggest that the factor of age could impose extreme effects on lipids and lipid-like molecules.

#### 3.3.2. Organic Acids and Derivatives

Following lipids and lipid-like molecules, a total of 53 organic acids and derivatives were detected, consisting of 46 amino acids, peptide and analogues, 5 hybrid peptides, 1 carboxylic acid derivative, and 1 tricarboxylic acid (Appendix A), suggesting that amino acids, peptide and analogues are the main compounds of organic acids and derivatives. Figure 4 illustrates the expression patterns of these 53 DMs from D150 to D450. In general, the expression patterns of these 53 DMs of organic acids were also more similar between D300 and D450, which is consistent with the results for free amino acids in this study (Table 1). These DMs could also be further classed into three distinct subclusters according to the expression pattern. Subcluster one (30 DMs), including balenine, anserine, L-beta-aspartyl-L-leucine, glutathione, oxidized glutathione, glycyl-lysine, alanyl-tyrosine, s-cysteinosuccinic acid, L-proline, L-tyrosine, L-arginine, and indicaxanthin, showed a lasting downregulation from D150 to D450. In addition, the contents of 12 DMs, clustered in subcluster two, such as carnosine, L-L-homoglutathione, vulgaxanthin II, alanyl-gamma-glutamate, cysteinyl-hydroxyproline, N-a-acetyl-L-arginine, N-acetyl-L-histidine, N-(1-deoxy-1-fructosyl)valine, 2-amino-6-hydroxyhexanoic acid, and L-histidine, were increased with age. In contrast, the contents of 11 DMs, clustered in subcluster three, including L-lysine, L-isoleucine, pyroglutamine, N-alpha-acetyllysine, N2-acetyl-L-ornithine, acetyl-DL-leucine, and N-acetyl-L-leucine, achieved the maximum at D300. This result indicates that amino acids, peptide and analogues, along with hybrid peptides, are greatly affected by age.

#### 3.3.3. Organooxygen Compounds and Flavonoids

In this study, 11 organooxygen compounds and 4 flavonoids were detected, respectively (Appendix A). The expression patterns of these 15 DMs from D150 to D450 are illustrated in Figure 5. These DMs could be further classed into three distinct subclusters according to the expression pattern. Subcluster one (5 DMs), including neohesperidin dihydrochalcone, demethyloleuropein, 7-chloro-2-(3,4-dimethoxyphenyl)-3,5-dihydroxy-6,8-dimethoxy-4H-chromen-4-one, and phenyl glucuronide, showed a steady upregulation from D150 to D450. In addition, the contents of 6-hydroxymelatonin glucuronide, epidermin, apigenin 7-sulfate, L-histidine, and sedoheptulose 7-phosphate, coming from subcluster two, showed down-regulated significantly from D150 to D300, then no significant change was detected from D300 to D450, while only one differential metabolite (2′-Hydroxyacetophenone) was detected and assigned to subcluster three, characterized by achieving maximum relative content at D300.

#### 3.3.4. Receiver Operating Characteristic Curve Analysis

The ROC curve was employed to evaluate the predictive performance of the aforementioned DMs based on D450 of BYC breast muscle. Some metabolites with AUC ≥ 0.95 are shown in Figure 6.

### 3.4. KEGG Pathway Enrichment and Topology Analysis

KEGG pathway enrichment and pathway topology analysis were employed to explore the biological pathways involved in DMs due to different ages. Based on the criteria of impact value of pathway impact more than 0.1, there were eight most impacted metabolic pathways between the comparison of D450 vs. D150, including glycerophospholipid metabolism, taurine and hypotaurine metabolism, glutathione metabolism, histidine metabolism, arginine and proline metabolism, tryptophan metabolism, arginine biosynthesis, tyrosine metabolism, whereas three metabolic pathways, including pantothenate and CoA biosynthesis, tryptophan metabolism, and lysine degradation were detected between D300 and D450 (Figure 7). These enriched pathways were mainly involved in the synthesis and metabolism of amino acid, peptide, and glycerophospholipid.

## 4. Discussion

### 4.1. Lipid Compounds

Prior studies have demonstrated a prominent number of phospholipids in the muscles and these play a central role in the formation of characteristic volatile flavors of meat products [20,21,22,23]. In addition, phospholipids are the key constituents of membrane involving in a broad range of cellular functions, including signal transduction and regulation of the transport process [24]. According to the results of this study, the main DM components classified as lipids and lipid-like molecules in BYC breast meat were PC (20 species) and PE (13 species). It was noteworthy that polyunsaturated fatty acids (PUFAs), such as docosapentaenoic acid (DPA), eicosapentaenoic acid (EPA), arachidonic acid (AA), and linoleic acid (LA), were not free fatty acids. They predominantly deposited in the glycerophospholipids instead. These results are consistent with previous studies focusing on Chinese indigenous chicken breeds, such as Taihe black-boned silky fowl, Guangyuan grey chicken, Tibetan chicken, and Jiuyuan black chicken [4,25]. Furthermore, the BYC breast meat at D450 had a significantly higher content of PUFA-enriched lipids (Appendix A), such as PC(18:2(9Z,12Z)/22:5(4Z,7Z,10Z,13Z,16Z)), PC(16:0/22:4(7Z,10Z,13Z,16Z)), PE(15:0/22:1(13Z)), PC(16:0/20:4(5Z,8Z,11Z,14Z)), PC(16:0/18:2(9Z,12Z)), PC(P-18:1(11Z)/16:0), which was consistent with our previous study, which reported that the breast meat of BYC at D450 presents a feature with a significantly higher concentration of PUFA as compared with that at D150 and D300 [15]. Notably, Lysoglycerophospholipid species (LGP) exhibited a lasting down-regulation trend from D150 to D450, except for LysoPE(18:1(11Z)/0:0) and LPC(18:3). However, PC- and PE-related glycerophospholipids exhibited the highest abundance at D450, except for PC(o-16:0/20:4(8Z,11Z,14Z,17Z)). This finding might be related to much higher levels of hydrolase and enzyme activation during the later growth process. Ge et al. [12] found that LysoPC(18:1), LysoPC(18:2), and LysoPC(16:0) in BYC breast meat increased from 56 days to 120 days, different from our results, which might be due to age differences. Altogether, these results suggested that having significantly higher PUFA-enriched glycerophospholipids of BYC at D450 was the root cause of the better properties of meat flavor and of health benefits.

Besides glycerophospholipids, some compounds of prenol lipids, such as ganoderic acid alpha and oleuroside, were also identified. Ganoderic acid alpha (GAA) is a distinguished bioactive compound belonging to the lanostane triterpenes, which presents a broad range of pharmacological attributes, such as anticancer, antioxidant, and anti-inflammatory [26,27]. GAA has been documented as possessing a hepatoprotective effect on liver injury and hepatic toxicity [28,29]. In this study, GAA was further identified with the highest VIP value (4.53) based on the OPLS-DA analysis (Figure 2), which could be considered as a prominent bioactive compound accounting for the nutritional and medicinal properties of BYC.

### 4.2. Peptide Compounds

Numerous experimental studies have demonstrated that peptides are widely regarded as components with special functional effects on flavor development, unique aromas and tastes during the thermal processing of foods [30,31]. Peptides containing glutamate or aspartate always present an umami taste in food [32]. In our study, besides the common amino acids, DMs classed as peptides were primarily identified, such as carnosine, anserine, balenine, L-beta-aspartyl-L-leucine, L-L-homoglutathione, glutathione (γ-L-glutamyl-L-cysteinyl-glycine; GSH), oxidized glutathione (GSSG) and cysteinyl-hydroxyproline. Of these, metabolites of L-beta-aspartyl-L-leucine, L-L-homoglutathione, GSH, and GSSG containing the residues of glutamyl or aspartyl, might be associated with the umami taste of BYC. In addition to an umami function, glutathione takes part in a series of physiological processes, including antioxidant function, regulation of cell cycle and cellular redox balance, a storage form, and a transport form of cysteine avoiding autoxidation [33]. In addition, GSH is involved in the formation of a sulfurous odor after heating [34].

Carnosine (β-alanyl-L-histidine) and its methylated analogs anserine (β-alanyl-1-methyl-L-histidine) and balenine (β-alanyl-3-methyl-L-histidine), known as histidine-containing imidazole dipeptides, have been demonstrated as accounting for the biological benefits of antioxidant, antifatigue, antiglycation and the buffering effect in muscle tissues [35,36,37]. It has been confirmed that histidine-containing imidazole dipeptides are present in chicken meat at high concentrations [38]. Kojima et al. [39] reported that Silky Fowl (*Gallus gallus dommesticus*) had 1.6- and 1.9-fold higher carnosine contents in thigh and breast meat compared with broilers. Moreover, dietary supplementation with meat with a high content of carnosine could prevent age-related diseases [40]. In this study, carnosine presented the highest at D450, followed by D300 and then D150, while anserine and balenine decreased with age, which was consistent with our results on meat quality characteristics (Table 1) and previous reports in Wuding chicken, Daheng broilers, and BYC [12,13,25]. All these data consistently confirm a decreasing ratio of anserine to carnosine in chicken with increasing age, which might be used as an excellent discriminator between chickens with different ages, and the relationship between this ratio and bird age requires further study. Taken together, in the case of the wide benefits of peptides as mentioned above, products of BYC could work as natural peptide sources to play a pivotal role in providing health benefits to consumers.

### 4.3. Flavonoid Compounds

It has been demonstrated that flavonoid compounds, with diverse biological presence, possess great health benefits for the human body, including antioxidant, antibacterial, anticancer, anti-inflammatory, anti-aging, and immunomodulatory activities [41,42]. During olive maturation, demethyloleuropein is reported to be a precursor for oleacein, which possesses a powerful antioxidant activity [43,44]. Neohesperidin dihydrochalcone, belonging to the citrus flavonoids, is characterized by its anti-inflammatory, antioxidant, and sweetness potential, and has been used against osteoporosis and osteoarthritis in bone health [45] and as a food additive (sweetener) [46]. Derivatives of chroman-4-one, classed as flavanones, have been applied to regulate cellular metabolism and scavenge free radicals [47,48]. Apigenin, belonging to the flavone subclass, is reported to halt cellular proliferation in human breast and liver cancer cells [49,50]. In this study, four flavonoids compounds were identified as DMs according to D450, including demethyloleuropein, neohesperidin dihydrochalcone, 7-chloro-2-(3,4-dimethoxyphenyl)-3,5-dihydroxy-6,8-dimethoxy-4H-chromen-4-one (a chromen-4-one derivative), and apigenin 7-sulfate (Appendix A). The concentration of demethyloleuropein, neohesperidin dihydrochalcone, and the chromen-4-one derivative showed an increasing trend throughout the developmental process of BYC, while the abundance of apigenin 7-sulfate declined from D150 to D300, and afterwards remained in a stable state from D300 to D450 (Figure 5). Overall, these results indicate that the abundance of flavonoid compounds exists at maximum at D450 in the breast meat of chicken, which accounts for the high nutritional value of BYC, particularly for chicken with older age.

## 5. Conclusions

In this study, the dynamic alterations of metabolite composition of BYC breast meat during the developmental process were evaluated by UHPLC-Q Exactive HF-X-based metabolomics and multivariate analysis. A total of 210 DMs were determined as the main discriminatory components throughout a wide age spectrum over 450 days. Carnosine, L-L-homoglutathione, chromen-4-one, neohesperidin dihydrochalcone, demethyloleuropein, and glycerophospholipids exhibited the highest abundance at D450. However, balenine, anserine, L-beta-aspartyl-L-leucine, glutathione, oxidized glutathione, ganoderic acid alpha, and Lysoglycerophospholipid species (LGP) exhibited a lasting down-regulation trend from D150 to D450. Additionally, AUC values of these most discriminant metabolites were further approved by AUC ≥ 0.95. A total of 10 pathways, including glycerophospholipid metabolism, glutathione metabolism, histidine metabolism, pantothenate and CoA biosynthesis, affecting the properties of meat flavor and health benefit through synthesis and metabolism compounds of glycerophospholipids, peptides, and flavonoids were further identified. Our study elucidates the alternations of metabolic profiling due to age and the potential bioactive compounds of BYC.

## Figures and Tables

**Figure 1 foods-13-02549-f001:**
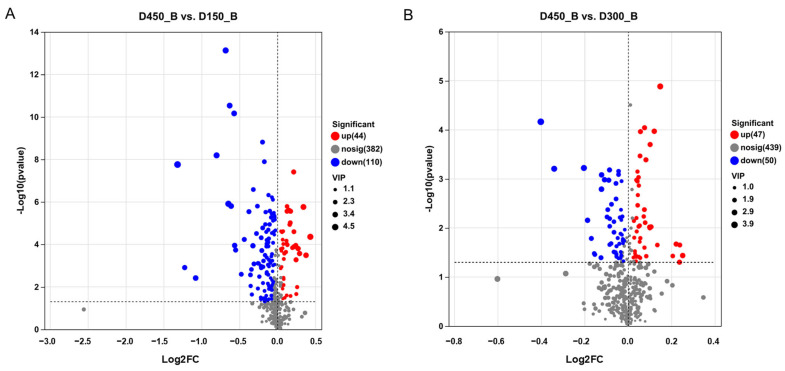
Volcano plot of *p* values between different ages according to D450 (D450 vs. D150, D450 vs. D300). The red and blue dots represent the significantly up-regulated and down-regulated metabolites, respectively, while gray dots represent no significantly differential metabolites. The dot size represents the value of variable importance in projection (VIP). (**A**) D450_B vs. D150_B; (**B**) D450_B vs. D300_B. D150_B: breast muscle at 150 days, D300_B: breast muscle at 300 days, D450_B: breast muscle at 450 days.

**Figure 2 foods-13-02549-f002:**
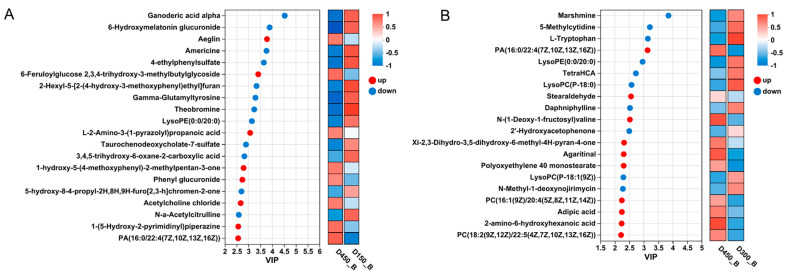
Variable importance in projection (VIP) scores based on the weighted coefficients of OPLS-DA model were applied to rank the top 20 DMs contributed to the metabolites discrimination between different ages according to D450. (**A**) D450_B vs. D150_B; (**B**) D450_B vs. D300_B. D150_B: breast muscle at 150 days, D300_B: breast muscle at 300 days, D450_B: breast muscle at 450 days.

**Figure 3 foods-13-02549-f003:**
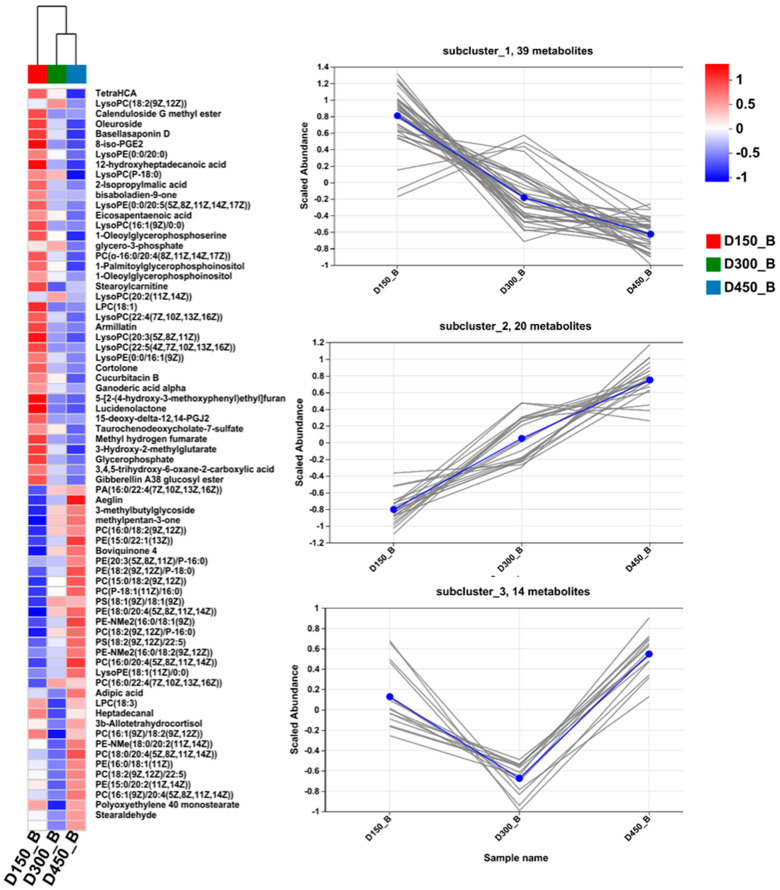
Heatmap (**left**) and subclustering (**right**) analysis of the expression patterns of the DMs of lipids and lipid-like molecules according to human metabolome database (HMDB) in breast muscle of Beijing-You chicken at three different age stages. D150_B: breast muscle at 150 days, D300_B: breast muscle at 300 days, D450_B: breast muscle at 450 days. Each gray line in the subclustering graph represents a differential metabolite, and the blue line represents the average expression level of all differential metabolites in the individual subcluster.

**Figure 4 foods-13-02549-f004:**
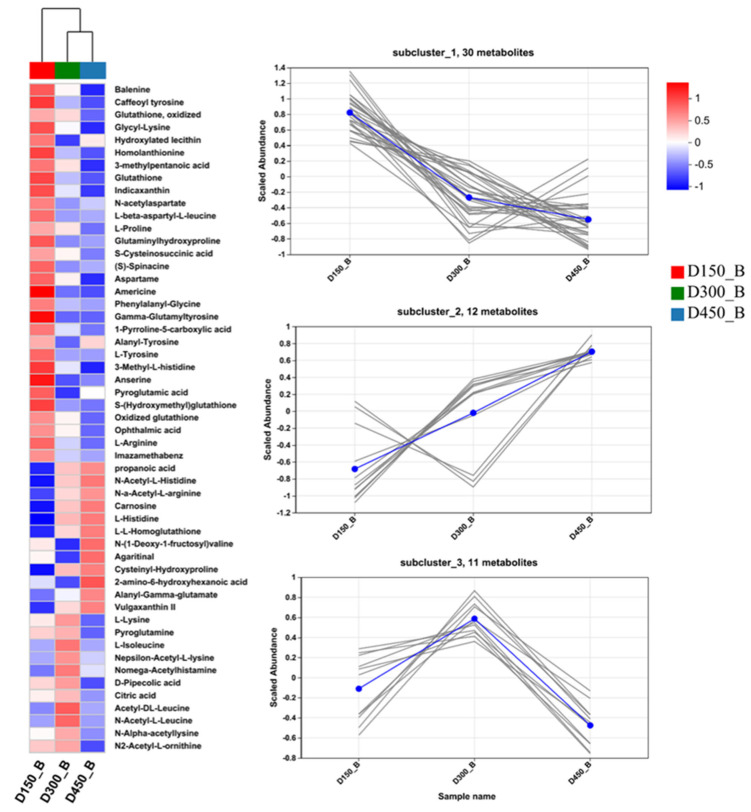
Heatmap (**left**) and subclustering (**right**) analysis of the expression patterns of the DMs of organic acids and derivatives based on human metabolome database (HMDB) in breast muscle of Beijing-You chicken at three different age stages. D150_B: breast muscle at 150 days, D300_B: breast muscle at 300 days, D450_B: breast muscle at 450 days. Each gray line in the subclustering graph represents a differential metabolite, and the blue line represents the average expression level of all differential metabolites in the individual subcluster.

**Figure 5 foods-13-02549-f005:**
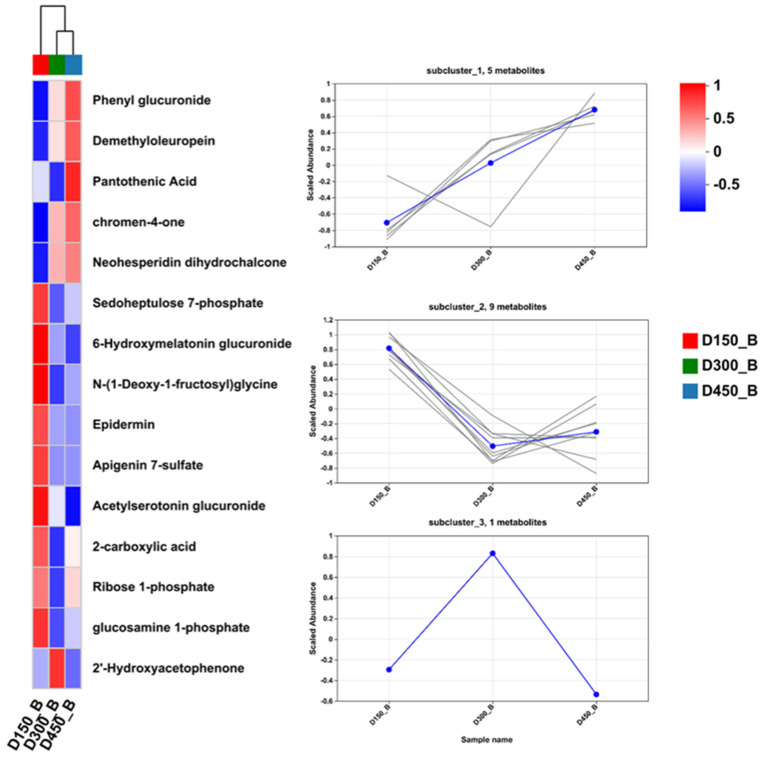
Heatmap (**left**) and subclustering (**right**) analysis of the expression patterns of the DMs of organooxygen compounds and flavonoids based on human metabolome database (HMDB) in breast muscle of Beijing-You chicken at three different age stages. D150_B: breast muscle at 150 days, D300_B: breast muscle at 300 days, D450_B: breast muscle at 450 days. Each gray line in the subclustering graph represents a differential metabolite, and the blue line represents the average expression level of all differential metabolites in the individual subcluster.

**Figure 6 foods-13-02549-f006:**
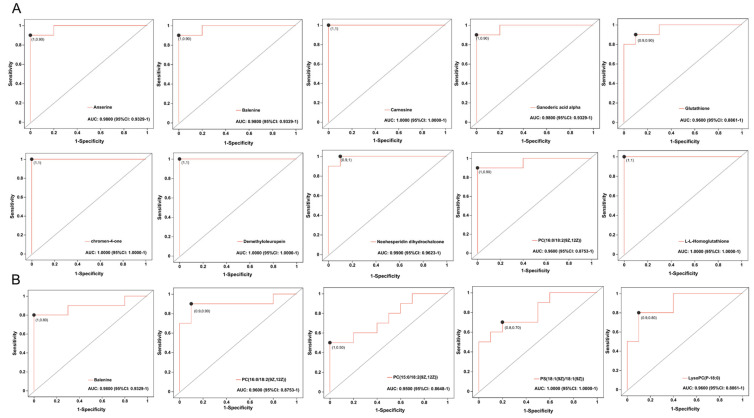
Receiver operating characteristic (ROC) curves for the metabolites between different ages comparison according to D450 with AUC ≥ 0.95. (**A**) D450 vs. D150; (**B**) D450 vs. D300. D150: day of 150; D300: day of 300; D450: day of 450.

**Figure 7 foods-13-02549-f007:**
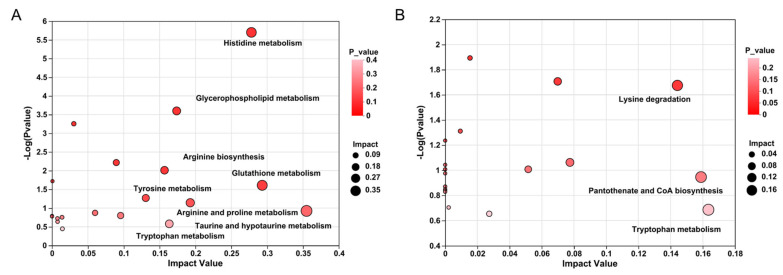
KEGG topology analysis of differential metabolites between different ages according to D450. (**A**) D450 vs. D150; (**B**) D450 vs. D300. The *X*-axis and *Y*-axis represent the pathway impact and pathway enrichment, respectively. The size and color of bubble stand for impact value and *p*-value, respectively. D150: day of 150; D300: day of 300; D450: day of 450.

**Table 1 foods-13-02549-t001:** Characteristics of chicken breast meat quality at three different ages (mean ± SD, *n* = 15) ^1^.

Traits ^2^	D150	D300	D450
live body weight (g)	1344 ± 63 ^c^	1951 ± 266 ^b^	2228 ± 294 ^a^
dry matter (%)	24.91 ± 0.99 ^c^	28.45 ± 0.97 ^b^	29.19 ± 1.02 ^a^
IMF (%)	1.36 ± 0.53 ^b^	3.10 ± 0.90 ^a^	3.25 ± 1.55 ^a^
Peptides			
Ans (mg/g)	9.54 ± 0.58 ^a^	7.58 ± 0.97 ^b^	6.77 ± 1.11 ^c^
Car (mg/g)	1.36 ± 0.74 ^c^	4.63 ± 1.21 ^b^	5.70 ± 1.12 ^a^
Free Amino Acids			
P-Ser (mg/g)	0.019 ± 0.004 ^a^	0.01 ± 0.003 ^b^	0.018 ± 0.005 ^a^
Tau (mg/g)	0.083 ± 0.021 ^a^	0.048 ± 0.007 ^b^	0.048 ± 0.007 ^b^
Asp (mg/g)	0.01 ± 0.013 ^b^	0.013 ± 0.005 ^b^	0.036 ± 0.012 ^a^
Thr (mg/g)	0.026 ± 0.01 ^b^	0.068 ± 0.052 ^a^	0.042 ± 0.015 ^b^
Ser (mg/g)	0.049 ± 0.012	0.056 ± 0.012	0.049 ± 0.006
Glu (mg/g)	0.061 ± 0.02 ^b^	0.085 ± 0.029 ^a^	0.07 ± 0.025 ^ab^
Gly (mg/g)	0.031 ± 0.009 ^c^	0.045 ± 0.013 ^a^	0.038 ± 0.007 ^b^
Ala (mg/g)	0.052 ± 0.016 ^c^	0.081 ± 0.02 ^a^	0.068 ± 0.013 ^b^
Val (mg/g)	0.018 ± 0.006 ^b^	0.037 ± 0.009 ^a^	0.03 ± 0.007 ^c^
Met (mg/g)	0.003 ± 0.001 ^b^	0.003 ± 0.002 ^b^	0.009 ± 0.002 ^a^
Ile (mg/g)	0.003 ± 0.001 ^b^	0.003 ± 0.002 ^b^	0.009 ± 0.002 ^a^
Leu (mg/g)	0.017 ± 0.003 ^b^	0.042 ± 0.01 ^a^	0.038 ± 0.008 ^a^
Tyr (mg/g)	0.016 ± 0.002 ^b^	0.021 ± 0.005 ^a^	0.022 ± 0.003 ^a^
Phe (mg/g)	0.011 ± 0.003 ^b^	0.019 ± 0.004 ^a^	0.02 ± 0.005 ^a^
b-Ala (mg/g)	0.052 ± 0.014 ^a^	0.024 ± 0.011 ^b^	0.023 ± 0.013 ^b^
Lys (mg/g)	0.032 ± 0.015 ^b^	0.047 ± 0.014 ^a^	0.033 ± 0.009 ^b^
His (mg/g)	0.007 ± 0.001 ^b^	0.019 ± 0.005 ^a^	0.017 ± 0.003 ^a^
Arg (mg/g)	0.024 ± 0.01 ^c^	0.052 ± 0.017 ^b^	0.065 ± 0.014 ^a^
Pro (mg/g)	0.021 ± 0.03	0.029 ± 0.005	0.022 ± 0.004
Total Amino Acids (mg/g)	0.535 ± 0.072 ^b^	0.701 ± 0.129 ^a^	0.657 ± 0.084 ^a^

^1^ Values within a row followed by different superscript letters (a–c) differ significantly (*p* ≤ 0.05). ^2^ IMF: intramuscular fat; Ans: anserine; Car: carnosine; P-Ser: O-Phosphoserine; Tau: taurine; Asp: aspartic acid; Thr: threonine; Ser: serine; Glu: glutamic acid; Gly: glycine; Ala: alanine; Val: valine; Met: methionine; Ile: isoleucine; Leu: leucine; Tyr: tyrosine; Phe: phenylalanine; b-Ala: β-alanine; Lys: lysine; His: histidine; Arg: arginine; Pro: proline. D150: day of 150; D300: day of 300; D450: day of 450.

## Data Availability

The original contributions presented in the study are included in the article/Appendix A, further inquiries can be directed to the corresponding author.

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
