# Peer review of "Metabolomics Reveals Glycerophospholipids, Peptides, and Flavonoids Contributing to Breast Meat Flavor and Benefit Properties of Beijing-You Chicken"

_foods, 2024, doi:10.3390/foods13162549_

Round 1

Reviewer 1 Report

Comments and Suggestions for Authors

The manuscript represents a good contribution to the field of meat science. However, several concerns must be considered by the authors in the revision step:

1) Introduction section must be revised by adding recent and appropriate references dealing with untargeted metabolomics in meat science.

2) Is the extraction step of the metabolites novel? Or any appropriate reference could be cited to support this important step?

3) How was prepared the pooled QC sample? 

4) Please, provide all the details related to ESI conditions and polarities of the Orbitrap mass spectrometer; also, what is the TopN of the instrument for DDA? Have you calibrated the instrument before running the sequence? Have you randomly injected blank samples (solvent only)?

5) No reference to the AA analysis (Table 1) can be found in M&M section.

6) Have the authors used OPLS-DA or PLS-DA? They are quite different approaches.

7) Have the authors validated their prediction model? For example, have the author carried out Hotelling's T2 test, permutation test, a Cross-Validation Anova on the residue? Have the authors calculated the ROC AUC values of the most discriminant metabolites? I do think that some very important data analyses are actually missing and/or not considered.

8) The resolution of some Figures must be improved (e.g., Figure 3, Figure 4)

9) How the authors explain the discriminant weight of phenolic compounds? The main findings have not been properly discussed from a food science & technology standpoint.

Comments on the Quality of English Language

English should be revised. Minor adjustments are required.

Reviewer 2 Report

Comments and Suggestions for Authors

General: There is a general assumption that consumer preference is directly correlated to bird age. L54 states generally that older birds are preferred by consumers vs younger birds, but refs 9 and 10 do not seem to report any data on consumer preference vs bird age. Furthermore, L59-60 suggests that studies on birds up to 450 days have not been thoroughly studied thus consumer preference of 450 days old birds is unclear. It is worth mentioning if there is an optimum age for harvest / consumption of Beijing You chickens, or if there is a certain age where the birds are considered “too old” based on consumer preferences. This will affect all subsequent analyses, especially identification of differential metabolites, since current analyses appears to be based on the assumption that D450 samples have the ideal meat quality.

L119-121: Please elaborate further on how identification was performed. Eg, which parameters were matched (MS? MS2?) what were the tolerances used? Was there any confirmation using authentic standards? This needs to be explained in greater detail for the untargeted metabolomics experiment.

L157: Correct Lye to Lys, also in table 1

Fig 3: Please increase font size for better legibility

Fig 4: Please increase font size for better legibility.

L329-332: Given that GAA is a potent bioactive compound which is depleted from day 150 to 450, wouldn’t that suggest that meat quality is poorer in older birds? This runs counter to the suggestion in L55-57 suggesting that older chickens have more health benefits.

L342-343: From Fig 4, GSH and GSSG are depleted from D150 to D450. Please clarify how the data supports the findings that umami taste increases with age in BYC.

Conclusions: Are the authors able to comment if these compounds may be entering / accumulating from the diets, based on feed used for example? Eg aspartame is found in the meat, but this is an artificial sweetener – thus may be possible that such compounds are entering via the diet.

Understanding feed composition may help to shed more light on the sources of these compounds rather than focus on the up/down regulation of metabolic pathways in chicken.

Round 2

Reviewer 1 Report

Comments and Suggestions for Authors

The manuscript has been improved and now the experimental design and M&M section is clearly presented. Final decision: Acceptance

Comments on the Quality of English Language

Minor improvements